# Does Routine Triple-Time-Point FDG PET/CT Imaging Improve the Detection of Liver Metastases?

**DOI:** 10.3390/diagnostics10090609

**Published:** 2020-08-19

**Authors:** Yu-An Yen, Wen-Sheng Huang, Chuang-Hsin Chiu, Yu-Chang Tyan, Jhi-Joung Wang, Li-Chun Wu, I Jung Feng, Chiang Hsuan Lee

**Affiliations:** 1Department of Nuclear Medicine, Chi Mei Medical Center, No.901, Zhonghua Rd., Yongkang Dist., Tainan City 710, Taiwan; yayena50803@gmail.com (Y.-A.Y.); zg0919@gmail.com (L.-C.W.); 2Department of Nuclear Medicine, Taipei Veterans General Hospital, No.201, Section 2, Shipai Rd., Beitou Dist., Taipei City 112, Taiwan; wshuang01@gmail.com; 3Department of Nuclear Medicine, National Defense Medical Center/Tri-Service General Hospital, No. 325, Sec. 2, Chenggong Rd., Neihu Dist., Taipei City 114, Taiwan; treasure316@gmail.com; 4Department of Medical Imaging and Radiological Sciences, Center for Infectious Disease and Cancer Research, Research Center for Environmental Medicine, Graduate Institute of Medicine, College of Medicine, Kaohsiung Medical University, Kaohsiung 807, Taiwan; yctyan@kmu.edu.tw; 5Institute of Medical Science and Technology, National Sun Yat-sen University, Kaohsiung 804, Taiwan; 6Department of Medical Research, Kaohsiung Medical University Hospital, Kaohsiung 807, Taiwan; 7Department of Medical Research, Chi Mei Medical Center, No.901, Zhonghua Rd., Yongkang Dist., Tainan City 710, Taiwan; 400002@mail.chimei.org.tw (J.-J.W.); ijungfeng112781@gmail.com (I.J.F.)

**Keywords:** dual-time-point, fluorodeoxyglucose, liver metastases, positron emission tomography/computed tomography, triple-time-point

## Abstract

Prior reports have demonstrated the improved ability of delayed fluorine-18 (^18^F) fluorodeoxyglucose (FDG) positron emission tomography/computed tomography (PET/CT) imaging (dual-time-point imaging) in detecting more patients with liver metastases. To evaluate whether routine triple-time-point FDG PET/CT imaging improves the detection of liver metastasis not visualized on initial imaging. To our knowledge, no triple-time-point imaging has been reported. This retrospective study included total 310 patients with various malignancies who underwent PET/CT scans. Triple-time-point imaging including the liver was obtained. The comparison between negative and positive liver lesions on delayed imaging for patients with initial negative imaging were analyzed. Of the 310 patients, 286 did not exhibit liver lesions on initial imaging, but six of the 286 patients exhibited lesions on delayed imaging. No additional liver lesions were detected on further delayed imaging in the 286 patients. The other 24 patients with liver lesions identified on initial imaging still showed lesions on delayed and further delayed imaging. The analysis showed a significant difference in the percentage of colorectal cancer (66.7%) and liver lesions before the PET scan (50.0%) compared with unchanged results (22.1% and 3.9%, respectively). Routine triple-time-point imaging did not improve the detection of liver metastases; however, it may be recommended in patients with colorectal cancer and liver lesions before the PET scan.

## 1. Introduction

The detection of liver metastases is critical for cancer staging and disease management. Therefore, our newly opened positron emission tomography (PET) center used the current PET/CT scan to emphasize the accuracy of PET results, especially in the detection of liver metastases.

Fluorine-18 (^18^F) fluorodeoxyglucose (FDG) PET/CT is highly sensitive and specific for identifying patients with liver metastases [1,2,3]. However, the relatively high uptake of the background in normal liver tissue may mask small or only mildly FDG-avid liver metastases [4].

The improved ability in detection of more patients with liver metastases with delayed FDG PET/CT imaging (dual-time-point imaging) has been reported [5,6,7,8,9,10,11,12,13]. The underlying rationale of dual-time-point imaging is the continued clearance of normal liver tissue activity or continued FDG accumulation in metastatic liver lesions. This effect often results in an improved image contrast of liver metastases [5,6,14,15].

Different time points are set for initial and delayed imaging, the mean time intervals ranging from 50 to 90 min and from 100 to 120 min after ^18^F-FDG administration [5,6,7,8,9,10,11,12,13]. There is currently no consensus as to what time delay is optimal for delayed imaging [16].

Taking advantage of previous reports in dual-time-point PET/CT imaging [5,6,7,8,9,10,11,12,13], we presumed that liver metastases may be detected only on delayed images, or even only on further delayed images. Therefore, further delayed imaging (triple-time-point imaging) was performed. This investigation focused on whether routine triple-time-point PET/CT imaging could improve the detection of liver metastases, especially in liver metastases detected only on delayed imaging, but not seen on initial imaging.

The current PET/CT scan owns a large field-of-view coverage and the highest NEMA sensitivity. The study was designed to assess liver metastases detected only seen on delayed rather than on initial FDG PET/CT imaging to evaluate, whether such an additional examination time and radiation exposure are worthy. To our knowledge, no triple-time-point imaging has been reported.

## 2. Material and Methods

### 2.1. Patients

This retrospective study was approved by the institutional review board of our hospital. The need for written informed consent was waived (assurance number: 10611-001, date of the approval: 10 May 2018). After excluding patients who were intolerant to longer examinations that did not involve delayed imaging, from March 2017 to April 2018, 310 patients with initial staging or the recurrence of various malignancies (120 women and 190 men; mean age, 58.2 years; range, 20–90 years) who underwent FDG PET/CT scans for initial staging or the recurrence of various malignancies were analyzed using a current PET/CT. No liver metastatic lesions were detected before the scans. Hence, no lesions were previously treated by interventional radiofrequency (RF) procedures. All patients with cancer were surveyed with at least one liver imaging modality (ultrasound (US) imaging or enhanced CT) before ^18^F FDG PET/CT imaging. Demographic characteristics of the 310 patients and their imaging findings are shown in Table 1 and Table 2.

### 2.2. Image Acquisition

We used the 16-section multidetector Discovery IQ PET/CT system with the five-ring configuration with a large, 26-cm field-of-view coverage and the highest NEMA (National Electrical Manufacturers Association) sensitivity in the industry, at up to 22 cps/kBq (GE Healthcare, Waukesha, WI, USA). The protocol of FDG PET/CT oncologic imaging started with low-dose CT without contrast medium from the vertex to the feet for attenuation correction of subsequent PET images and anatomic localization at approximately 60 min after the FDG injection. CT images were acquired using the 16-section multidetector scanner with an automatic exposure control system, performing 3D mA modulation according to the patient’s body size and attenuation level, ranging from 15 to 50 mA; the tube voltage was 120 kVp. Initial PET acquisition was performed in the same region, acquisition type 3D, 2 min/bed position from the vertex to the upper thigh and 1 min/bed position from the upper thigh to the feet [17]; ordered subset expectation maximization reconstruction settings: filter cut-off 6.4 mm, four iterations, and twelve subsets.

Then, delayed limited-area imaging including the liver was performed 100–120 min (mean 109 min) after the 370 ± 74 MBq FDG injection, and further delayed imaging focused on the liver was performed at 140–160 min (mean 151 min) with the same CT and PET parameters as the initial imaging, except for the 3- and 5-min/bed position on delayed and further delayed imaging for PET acquisition, respectively. In general, delayed imaging was acquired only in selected bodily regions based on patient history and findings on initial imaging, but had to include the liver, and further delayed imaging included only the liver.

### 2.3. Image Evaluation

Two experienced nuclear medicine physicians with knowledge of prior findings detected liver lesions by visual interpretation in consensus using a Xeleris 3.0 workstation (GE Medical Systems, Chicago, IL, USA), separately. In case of disagreement, the cases were discussed and consensus was obtained.

For proper interpretation, a thorough knowledge of the patient’s history was mandatory. Comparisons with other recent imaging modalities, such as CT, MRI, or ultrasound imaging, were performed for the diagnosis of the lesion, especially for the liver. If liver lesions were detected by any imaging modality including PET/CT, a final diagnosis of the presence of liver metastases or not would be pathologically confirmed if feasible under the patients’ informed consent, or would be correlated with the other liver imaging modalities. Enhanced CT has better diagnostic performance than US in the detection of liver metastases. Hence, we performed enhanced CT regardless of whether the detection of liver metastases on US was possible before the PET scan or not. MRI has higher sensitivity in detecting liver metastases than CT. Therefore, even if enhanced CT does not detect the metastatic liver lesions, they could be identified by enhanced MRI with Gadolinium or Primovist, a hepatospecific contrast medium (a self-paid item in our hospital). Alternatively, if no liver metastases were detected on enhanced CT before performing the PET scan, we reviewed the CT images. If CT images still yielded negative results, we then resorted to enhanced MRI. Therefore, enhanced CT or MRI served as the reference and gold standard in the detection of liver metastases in this research. Adequate management was adopted by the clinical physician. If no liver lesions were detected by any of the imaging modalities, the final diagnosis was no liver metastases.

### 2.4. Statistical Analysis

Patients with FDG-avid liver lesions detected only on delayed imaging, but not on initial imaging were the focus of this study (six patients in this study). The liver maximum standardized uptake value (SUV_max_) and tumor-to-normal liver ratio (TNR; defined as the SUV_max_ of the liver lesion to the mean standardized uptake value of the normal liver tissue ratio in this study) were calculated and correlated with the visual interpretation. The SUV_max_ of the liver lesion on initial imaging was measured in the same region as that on delayed imaging.

For the patients without liver lesion on initial imaging, we further compared only the clinical characteristics between negative- and positive-PET liver lesions on the delayed imaging to identify the differences. For the explored variables, age, gender, type of tumor, initial or recurrent cancer staging before the PET scan, and the liver lesion on other imaging modalities before the PET scan were presented separately. Due to the distribution of the collected sample, the comparisons of variables between two groups were analyzed by Fisher’s exact test for categorical variables and the Mann–Whitney U test for continuous variables.

A Friedman test was used to assess the liver lesion visible only on delayed imaging, but not on initial imaging whether there is a statistical difference of SUV_max_ in initial, delayed, and further delayed imaging.

A *p*-value of less than 0.05 was considered statistically significant in this study. Statistical analyses were conducted using Statistical Analysis Software (version 9.4, SAS Institute, Cary, NC, USA).

## 3. Results

Among the 310 patients, all 24 patients with liver lesions (10 patients had one lesion; others had multiple lesions) identified on initial imaging were still clearly seen on delayed and further delayed imaging, and there were 286 patients with no liver lesion on initial imaging (Figure 1). Of the 286 patients, six (2.1%) had liver lesions (each patient had one lesion) that were detected only on delayed imaging and more evident on further delayed imaging (i.e., rising TNRs with time) (Figure 2 and Figure 3). No additional liver lesions appeared on further delayed imaging in these 286 (six and the other 280) patients (Figure 1). All six patients with liver lesions detected on delayed imaging showed the same areas and shape as seen on further delayed imaging. Liver metastatic lesions were subsequently identified by pathology or radiology and critically changed cancer stage and disease management. These patients represented 4 out of 40 patients with colorectal cancer, 1 out of 20 patients with double cancer, and 1 out of 53 patients with head and neck cancer.

According to the result of the Friedman test, no significant difference of the SUV_max_ values among initial, delayed, and further delayed imaging in these six patients (Figure 4). However, the corresponding TNRs among these images were significantly more increased (Figure 5), and were consistent with the visual interpretation (Figure 2 and Figure 3).

As shown in Table 2, patients with colorectal cancer were separately found in 66.7% (4/6) of the group with positive delayed imaging and in 22.1% (62/280) of the group with negative delayed imaging. Fifty percent (3/6) patients in the group with positive delayed imaging and 3.9% (11/ 280) patients in the group with negative delayed imaging were identified with liver lesions on other imaging modalities before the PET scan. Both groups were significantly different from one another. Eleven patients had liver lesions detected only by the other modalities (not by PET), but were all interpreted as benign lesions (such as hemangioma or cyst).

Six patients with liver lesions detected only on delayed imaging, and 24 patients with liver lesions identified during the initial imaging were still clearly seen on delayed images. Thus, a total of 30 patients with liver metastases were detected in the cohort study. The detection sensitivity, specificity, positive predictive value, negative predictive value and accuracy for tests with negative liver initial imaging and positive delayed imaging are shown in Table 3. Two groups were analyzed using the Friedman test for evaluating patients with liver lesions only on delayed imaging, but not visible on initial imaging and delayed imaging. No further comparisons between the delayed and further delayed images were made because of absence of any lesions on further delayed imaging. Initial imaging had 80% detection sensitivity, 100% specificity, 100% positive predictive value, 97.9% negative predictive value and 98.1% accuracy. However, delayed imaging and further delayed imaging all had 100% detection sensitivity, specificity, positive predictive value, negative predictive value and accuracy, respectively.

The major advantage of delayed imaging is the increased detection sensitivity (80% vs. 100%), negative predictive value (97.9% vs. 100%), and accuracy (98.1% vs. 100%).

## 4. Discussion

As per our preliminary data, six of the 286 patients (2.1%) had liver metastases identified only on delayed PET/CT imaging. The mere 2.1% of patients identified may not indicate the importance of delayed imaging, but when it detects liver metastases, it still plays a crucial role in individual disease management. Therefore, the value of delayed imaging should not be understated. The other 24 patients with liver lesions identified on initial imaging were still seen on delayed and further delayed images. Similar to a prior review article [16], such images will generally not provide additional value if the diagnosis can be established confidently on initial imaging.

It may not be cost-effective or practical to perform delayed PET imaging, or even further delayed images, on every patient in PET centers with high patient throughput, and it may be more feasible to consider this approach in selected patients who have a higher likelihood of lesion detection only on delayed imaging. According to the findings in Table 2, patients with negative initial liver imaging but positive delayed imaging showed a different proportion of colorectal cancer and liver lesions on the other imaging modalities before the PET scan. This finding was not surprising, because colorectal cancers are known to most commonly metastasize to the liver, and those with a liver lesion already seen on the other imaging modalities (solid nodules on abdominal echo or enhancement nodules on CT) prior to the PET scan are generally regarded as having a high pretest probability of metastases.

The SUV_max_ values of these six patients with liver metastases identified only on delayed PET/CT imaging did not exhibit a continuous increase with time, but the corresponding TNRs continued to increase with time (Figure 4 and Figure 5), which was consistent with the visual interpretation (Figure 2 and Figure 3). The findings in were in agreement with the notion that background activity generally decreases on delayed imaging, leading to improved imaging quality rather than a continuously accumulating neoplastic FDG uptake. However, more data are required for further clarification.

The major advantage of delayed imaging is the increased detection sensitivity (80% vs. 100%), negative predictive value (97.9% vs. 100%), and accuracy (98.1% vs. 100%); therefore, this imaging technology should be used in clinical practice to optimize diagnostic performance [5,6,7,8,9,10,11,12,13,16]. In the present investigation, delayed imaging (dual-time-point imaging), including the liver, appeared to be enough to detect liver metastases, and further delayed imaging (triple-time-point imaging) was thus not necessary. However, further delayed images might still be beneficial for showing liver lesions in the same areas with the same shape, as delayed images provide us more confidence that the lesions are not caused by noise or artifacts (Figure 2 and Figure 3), with the increased contrast against the normal liver tissue (i.e., confirming liver lesions) and showing a long-term change in TNR, whereby if it continues rising, it might indicate a tendency towards malignant, rather than benign lesions [18,19,20,21] (i.e., diagnostic malignancy). Therefore, further delayed imaging may be useful for either confirming or excluding the presence of liver metastases with increased diagnostic performance and interpreter confidence.

The application of further delayed liver imaging results in more radiation exposure to patients, but we used a current low-dose CT with a new technique [16] (i.e., a new automatic exposure control system and a new reconstruction) and acquired only one bed (26-cm field of view), thereby limiting additional radiation exposure (when the tube current was 15 mA, DLP was 20.10 mGy-cm and was equal to 0.29 mSv, according to the dose report in the Xeleris workstation). Taking more time for further delayed imaging of the liver might be a disadvantage. However, the current PET/CT scan needed only one bed due to the larger field of view and increased acquiring speed due to high NEMA sensitivity, resulting in reduced imaging times (5 min for one bed) by approximately a quarter when compared to the old style PET/CT machine (two beds and 6–8 min/bed position, i.e., 12–16 min/bed position in total) to obtain a delayed liver image. Considering the pros and cons of this technique, the further delayed imaging in the liver might be recommended for the detection of liver metastases in patients with colorectal cancer or known liver lesions before the PET scan. Given the value of precision medicine, the cost effectiveness of this technology should also be considered an advantage. Current PET/CT with high NEMA sensitivity and an acquisition time set at 5 min/bed position for further delayed imaging were still enough to maintain image quality (Figure 2 and Figure 3).

We used a triple-time-point protocol for our routine protocol at initial setup to see whether it could detect more patients with liver metastases. However, after this retrospective study, we changed our routine protocol to a dual-time-point protocol, except in patients with colorectal cancer and with liver lesions before the PET scan.

In conclusion, as routine triple-time-point PET/CT imaging did not improve the detection of liver metastases, it can be recommended that in patients with colorectal cancer and with liver lesions before the PET scan, the use of a current PET/CT should be regarded as the most practical and cost-effective approach. Triple-time-point PET/CT imaging could confirm liver lesions and indicate the tendency of malignancy. However, more data are required for further clarifications in the future.

## Figures and Tables

**Figure 1 diagnostics-10-00609-f001:**
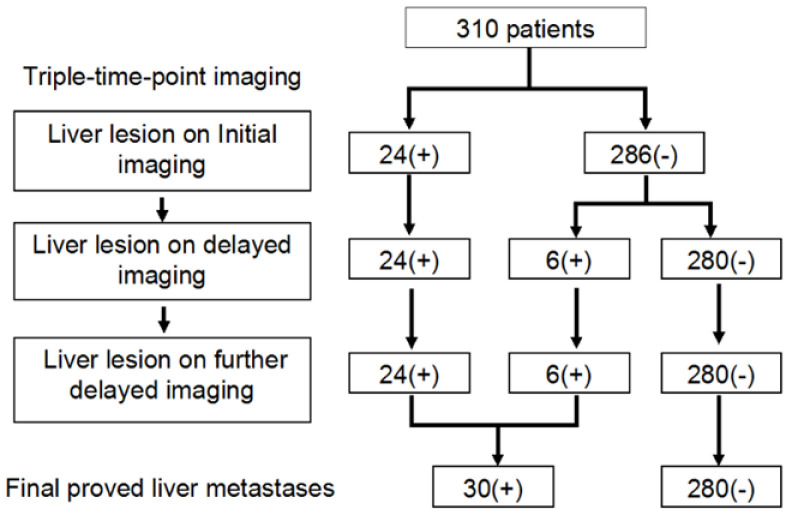
Flow chart demonstrating enrolled patients and their reporting. Of the total 310 patients, 286 patients had negative initial liver PET/CT imaging, six of the 286 patients had liver lesions on delayed imaging and further delayed imaging, and the other 24 patients exhibited liver lesions on initial imaging as well as delayed imaging. A total of 30 patients had liver metastases.

**Figure 2 diagnostics-10-00609-f002:**
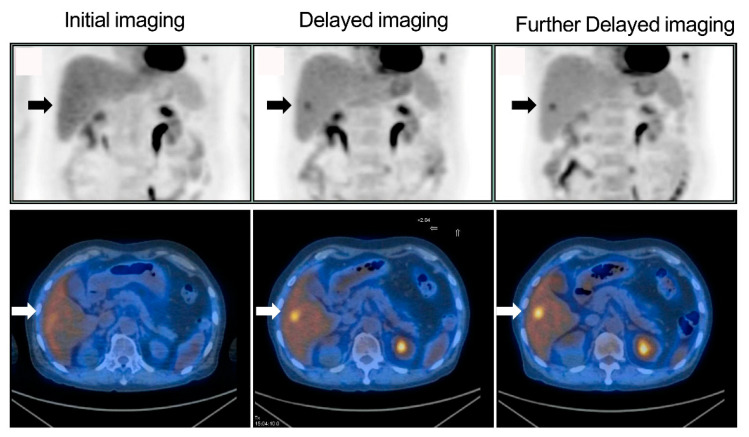
Liver metastases were detected only on delayed FDG PET/CT imaging. Liver FDG maximum-intensity projection (top row) and axial FDG PET/CT imaging (bottom row) of initial (left), delayed (middle) and further delayed imaging (right) in a 62-year-old male patient with nasopharyngeal cancer (NPC) displayed an FDG-avid liver lesion that was detected only on delayed and further delayed imaging but not on initial imaging (arrows). There was slightly more evidence of the liver lesion and increased contrast against normal liver tissue on further delayed imaging as compared with delayed imaging. These images were typical of a liver lesion detected only on delayed FDG PET/CT imaging but not on initial imaging. Pathology confirmed metastatic carcinoma of NPC origin.

**Figure 3 diagnostics-10-00609-f003:**
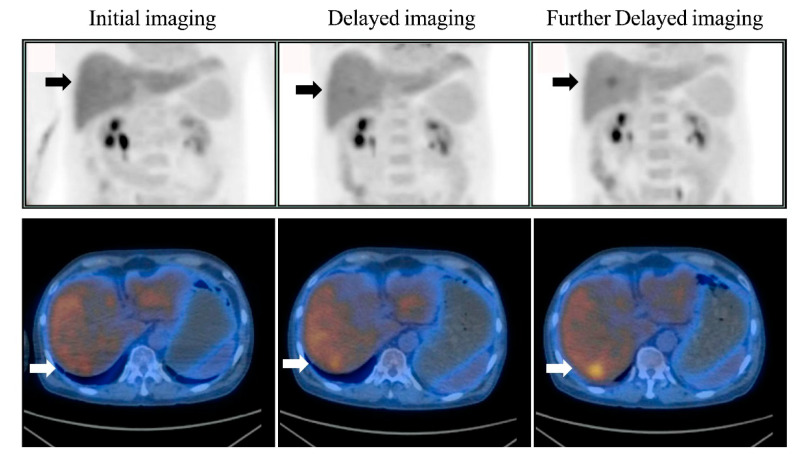
Liver metastases were detected only on delayed FDG PET/CT imaging. Liver FDG maximum-intensity projection (top row) and axial FDG PET/CT imaging (bottom row) of initial (left), delayed (middle) and further delayed imaging (right) in a 60-year-old-male patient with colon cancer displayed an FDG-avid liver lesion that was detected only on delayed and further delayed imaging but not on initial imaging (arrows). There was more evidence of the liver lesion and increased contrast against normal liver tissue on further delayed imaging as compared with delayed imaging. Pathology confirmed metastatic adenocarcinoma originating from the colon.

**Figure 4 diagnostics-10-00609-f004:**
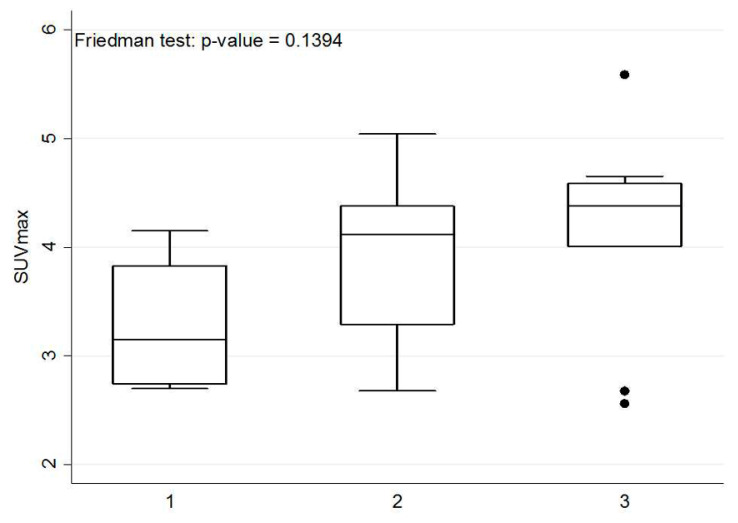
Box plot of SUV_max_ on initial, delayed and further delayed PET/CT imaging. The x-axis represents three separate images: initial imaging = 1; delayed imaging = 2; further delayed imaging = 3. Friedman test: *p*-value = 0.1394. The SUV_max_ values did not continue to increase with time and were not statistically significant.

**Figure 5 diagnostics-10-00609-f005:**
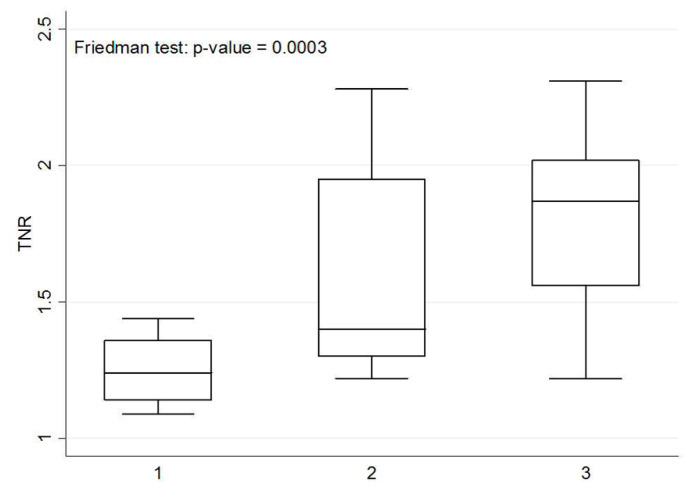
Box plot of TNR based on initial, delayed and further delayed PET/CT imaging. The x-axis represents three separate images: initial image = 1; delayed image = 2; further delayed image = 3. Friedman test: *p*-value = 0.0003. The TNRs appeared to increase with time and were identified to be significantly different.

**Table 1 diagnostics-10-00609-t001:** Patient characteristics.

Characteristic	Data	Positive Numbers Only on Delayed PET Imaging
**Patients (*n*)**	310	
**Mean age (y)**	58.2 (range, 20–90)	
Male gender (*n*)	190 (61.3%)	
Female gender (*n*)	120 (38.7%)	
**Cancer types (*n*)**		
Lymphoma	61 (19.7%)	
Head and neck cancer	53 (17.1%)	1
Esophageal cancer	38 (12.3%)	
Lung cancer	36 (11.6%)	
Breast cancer	30 (9.7%)	
Rectal cancer	23 (7.4%)	2
Double cancer	20 (6.5%)	1
Colon cancer	17 (5.5%)	2
Cervical cancer	12 (3.8%)	
Thyroid cancer	10 (3.2%)	
Melanoma	7 (2.3%)	
Cholangiocarcinoma	1 (0.3%)	
Pancreatic cancer	1 (0.3%)	
Endometrial cancer	1 (0.3%)	

**Table 2 diagnostics-10-00609-t002:** Comparison of clinical characteristics of negative and positive liver PET/CT findings on delayed imaging of 286 patients with negative initial liver PET/CT.

Initial Imaging	Delayed Imaging		
(−), *N* = 286	(−), *n* = 280	(+), *n* = 6	*p*-Value
**Mean age (y)**	58.0	61.0	0.4754
**Gender, *n* (%)**			0.0847
Male	168 (60.0)	6 (100.0)	
Female	112 (40.0)	0 (0.0)	
**Type of cancer, *n* (%)**			0.0269 *
Colon and rectum	62 (22.1)	4 (66.7)	
Non-colon or rectum	218 (77.9)	2 (33.3)	
**Purpose of PET**			
Initial staging, *n* (%)	103 (36.8)	0 (0.0)	0.0908
Recurrence, *n* (%)	177 (63.2)	6 (100.0)	0.0908
**Known tumor staging before the PET scan, *n* (%)**			0.4435
Stage 0–II	113 (40.4)	1 (16.7)	
Stage III–IV	164 (58.8)	5 (83.3)	
Unknown	3 (1.1)	0 (0.0)	
**Liver lesion on other images before the PET scan**			0.0017 *
**(+)**	11 (3.9)	3 (50.0)	
**(−)**	269 (96.1)	3 (50.0)	

* A *p*-value < 0.05 was considered statistically significant. Two groups were analyzed by Fisher’s exact test for categorical variables and the Mann–Whitney U test was used for continuous variables.

**Table 3 diagnostics-10-00609-t003:** Detection sensitivity, specificity, positive predictive value (PPV), negative predictive value (NPV) and accuracy for testing of only initial negative liver PET/CT images and delayed positive delayed PET/CT images.

Liver *n* (%)	Initial Imaging	Delayed Imaging	Further Delayed Imaging	*p*-Value
Sensitivity	24/30 (80.0)	30/30 (100.0)	30/30 (100.0)	0.0099
Specificity	280/280 (100.0)	280/280 (100.0)	280/280 (100.0)	-
PPV	24/24 (100.0)	30/30 (100.0)	30/30 (100.0)	-
NPV	280/286 (97.9)	280/280 (100.0)	280/280 (100.0)	0.0146 *
Accuracy	304/310 (98.1)	310/310 (100.0)	310/310 (100.0)	0.0139

Note: The major advantage of delayed imaging is the increased detection sensitivity (80% vs. 100%), negative predictive value (97.9% vs. 100%), and accuracy (98.1% vs. 100%). Delayed imaging was enough to detect liver metastases, and a further delayed imaging was thus not necessary. * A *p*-value < 0.05 was considered statistically significant in this study. Two groups were analyzed by the Friedman test for evaluating liver lesions only on delayed imaging, but not visible on initial imaging and delayed imaging. The further delayed imaging group was not analyzed, because there were no changes between the delayed and further delayed imaging groups.

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
