# Peer review of "Does Routine Triple-Time-Point FDG PET/CT Imaging Improve the Detection of Liver Metastases?"

_diagnostics, 2020, doi:10.3390/diagnostics10090609_

Round 1
Reviewer 1 Report
This is a well written interesting paper.
Abstract: adequate
Introduction: The Authors summarize the role of FDG PET CT in the detection of liver metastases, mainly reporting the different time points of dual-time-point. I think could be useful, for readers, to add some data about diagnostic accuracy reported in literature.
You have reported the following concept: "We used triple-time-point protocol for our routine protocol at initial setup to see whether it can detect more patients with liver metastases. However, after this retrospective study, we changed our routine protocol to dual-time-point protocol, except in patients with colorectal cancer and with liver lesions before the PET scan".
I suggest to move into a different part of the manuscript this sentence: it could be only reported in the conclusion or in the discussion (it sounds like a conclusion, based on results found in your analysis).
Materials and methods: Well written. Other radiological procedures performed for staging disease should be specified, if possible. Namely, which imaging modality was considered the reference for the analysis of lesion detection? You have reported that all imaging modalities were used for the diagnosis of metastases, but it is not clear which consideration was made in cases of discrepancy among imaging modalities (CT and/or MRI and/or PET).
Please, specify if among other imaging modalities used for the final assessment of lesions, you have included also MRI using hepatospecific contrast medium.
No comments for statistical analysis.
Results and discussion: median dose for dual-time-point and triple-time point should be reported. A critical comparison for doses between the 2 different time points could be improved, based on median values reported.
All six hepatic lesions positive depicted on delayed scan were found in recurrence: no lesions were previously treated by interventional RF procedures?
Conclusion, references and figures: adequate.
Author Response
Point 1: Introduction: The Authors summarize the role of FDG PET CT in the detection of liver metastases, mainly reporting the different time points of dual-time-point. I think could be useful, for readers, to add some data about diagnostic accuracy reported in literature.
You have reported the following concept: "We used triple-time-point protocol for our routine protocol at initial setup to see whether it can detect more patients with liver metastases. However, after this retrospective study, we changed our routine protocol to dual-time-point protocol, except in patients with colorectal cancer and with liver lesions before the PET scan".
I suggest to move into a different part of the manuscript this sentence: it could be only reported in the conclusion or in the discussion (it sounds like a conclusion, based on results found in your analysis).
Response 1:
According to your comment, we have moved the following sentence to the final part of the Discussion section (because no Conclusion section was included in this article): “We used triple-time-point protocol for our routine protocol at initial setup to see whether it can detect more patients with liver metastases. However, after this retrospective study, we changed our routine protocol to dual-time-point protocol, except in patients with colorectal cancer and with liver lesions before the PET scan.” (4. Discussion section, lines 281–284, page 7).
Point 2: Materials and methods: Well written. Other radiological procedures performed for staging disease should be specified, if possible.
Response 2
We added “ultrasound (US) imaging or enhanced CT” to the sentence as follows: All patients with cancer are surveyed with at least one liver imaging modality (ultrasound (US) imaging or enhanced CT) before 18F FDG PET/CT imaging (2. Materials and Methods section, line 82, page 2).
Point 3: Namely, which imaging modality was considered the reference for the analysis of lesion detection? You have reported that all imaging modalities were used for the diagnosis of metastases, but it is not clear which consideration was made in cases of discrepancy among imaging modalities (CT and/or MRI and/or PET). Please, specify if among other imaging modalities used for the final assessment of lesions, you have included also MRI using hepatospecific contrast medium.
Response 3
As per your comment, we added the sentence “Enhanced CT shows better diagnostic performance than US in detecting liver metastases.” Therefore, we performed enhanced CT regardless of whether the detection of liver metastases on US was possible before the PET scan or not. MRI has higher sensitivity for detecting liver metastases than CT. Therefore, even if enhanced CT does not detect metastatic liver lesions, they could be identified by enhanced MRI with Gadolinium or Primovist, a hepatospecific contrast medium (a self-paid item in our hospital). Conversely, if no liver metastases were detected on enhanced CT prior to performing the PET scan, we reviewed the CT images. If the CT images still yielded negative results, we then resorted to enhanced MRI. Therefore, enhanced CT or MRI served as the reference and gold standard for the detection of liver metastases in this investigation (2. Materials and Methods section, line 119-128, page 4).
No comments for statistical analysis.
Point 4: Results and discussion: median dose for dual-time-point and triple-time point should be reported. A critical comparison for doses between the 2 different time points could be improved, based on median values reported.
Response 4:
We obtained dual-time-point and further delayed images (triple-time-point images) for all participants. Median dose remained the same in both methods. We added the median dose “370 ± 74 MBq” to the sentence as follows: “Then, delayed limited-area imaging including the liver was performed 100–120 min (mean 109 min) after the 370 ± 74 MBq FDG injection, and further delayed imaging focused on the liver was performed at 140–160 min (mean 151 min) with the same CT and PET parameters as the initial imaging, except for the 3- and 5-min/bed position on delayed and further delayed imaging for PET acquisition, respectively.” (2. Materials and Methods section, line 104, page 3).
Point 5: All six hepatic lesions positive depicted on delayed scan were found in recurrence: no lesions were previously treated by interventional RF procedures?
Response 5
Since no lesions were detected before the scans, all patients were in the initial or recurrence stage of various malignancies during the routine survey. Therefore, no lesions were previously treated by interventional RF procedures. We added the sentence: “No liver metastatic lesions were detected before the scans. Hence, no lesions were previously treated by interventional radiofrequency (RF) procedures” (2. Materials and Methods section, line 80-81, page 2).
Conclusion, references and figures: adequate.

Reviewer 2 Report
In their original article titled 'does Routine triple-time-Point FDG PET/CT Imaging improve the detection of liver metastases' Yen et al retrospectively analysed FDG PET Images of 310 Cancer patients referred for staging / restaging. 30/310 patients were found to have liver metastases on PET out of which 6 were only detected on delayed Images. Out of These 6 patients, 4 (66.7%) patients had colorectal cancer. Based on the results authors concluded that triple time point imaging does not lead to any significant improvement in the detection of liver metastases except in colorectal Cancer.
The only major strength of the study is large number of patients. otehrwise, the Information contained in the manuscript is not really new. Apart from that authors have not mentioned which other Imaging method were used for the detection of liver metastases. CT has lower sensitivity as compared to MRI. MRI has higher sensitivity as compared to PET. These Imaging modalities cannot be used interchangeably, specifically for a study like this in which Gold Standard is the mentioned as 'other Imaging method'. These limitations have to be mentioned in detail.
The references used are not uptodate. there are several new studies in pubmed (36 in 2019-2020 if one searches with the words dual Point + FDG PET)
Author Response
Point 1: The only major strength of the study is large number of patients. otehrwise, the Information contained in the manuscript is not really new. Apart from that authors have not mentioned which other Imaging method were used for the detection of liver metastases. CT has lower sensitivity as compared to MRI. MRI has higher sensitivity as compared to PET. These Imaging modalities cannot be used interchangeably, specifically for a study like this in which Gold Standard is the mentioned as 'other Imaging method'. These limitations have to be mentioned in detail.
Response 1
According to your comment, we added the sentence: Enhanced CT shows better diagnostic performance than US in the detection of liver metastases. Therefore, we conducted enhanced CT regardless of whether the detection of liver metastases on US was possible prior to the PET scan or not. In addition, MRI has higher sensitivity for detecting liver metastases than CT. Therefore, even if enhanced CT does not detect the metastatic liver lesions, they could be identified by enhanced MRI with Gadolinium or Primovist, a hepatospecific contrast medium (a self-paid item at our hospital). Conversely, we reviewed the CT images if no metastatic liver lesions were detected on enhanced CT prior to the PET scan. Enhanced MRI was performed if the CT images were still negative. Therefore, enhanced CT or MRI was the reference and gold standard for the detection of liver metastases in this investigation. (2. Materials and Methods section, line 119-128, page 4)
Point 2: The references used are not uptodate. there are several new studies in pubmed (36 in 2019-2020 if one searches with the words dual Point + FDG PET)
Response 2:
We conducted a search using the words “dual images + FDG PET + liver metastases” and excluded not associated with liver metastases in 2018–2020. We were able to access three papers and added them to the references (References section, lines 327-334, page 9).
